# PTH-Induced Bone Regeneration and Vascular Modulation Are Both Dependent on Endothelial Signaling

**DOI:** 10.3390/cells11050897

**Published:** 2022-03-05

**Authors:** Doron Cohn-Schwartz, Yeshai Schary, Eran Yalon, Zoe Krut, Xiaoyu Da, Edward M. Schwarz, Dan Gazit, Gadi Pelled, Zulma Gazit

**Affiliations:** 1Department of Internal Medicine B, Division of Internal Medicine, Rambam Healthcare Campus, Haifa 3109601, Israel; d_cohnschwartz@rambam.health.gov.il; 2Skeletal Biotech Laboratory, Faculty of Dental Medicine, The Hebrew University of Jerusalem, Jerusalem 91120, Israel; yeshaischary@mail.tau.ac.il (Y.S.); eran.yalon@mail.huji.ac.il (E.Y.); dan.gazit@csmc.edu (D.G.); gadi.pelled@cshs.org (G.P.); 3Department of Surgery, Cedars-Sinai Medical Center, Los Angeles, CA 90048, USA; zoe.krut@cshs.org; 4Board of Governors Regenerative Medicine Institute, Cedars-Sinai Medical Center, Los Angeles, CA 90048, USA; 5Biomedical Imaging Research Institute, Cedars-Sinai Medical Center, Los Angeles, CA 90048, USA; xiaoyu.da@cshs.org; 6The Center for Musculoskeletal Research, Department of Orthopaedics, School of Medicine & Dentistry, University of Rochester, Rochester, NY 14642, USA; edward_schwarz@urmc.rochester.edu; 7Department of Orthopedics, Cedars-Sinai Medical Center, Los Angeles, CA 90048, USA

**Keywords:** parathyroid hormone, fracture healing, angiogenesis, osteogenesis, allograft, calvarial bone repair

## Abstract

The use of a bone allograft presents a promising approach for healing nonunion fractures. We have previously reported that parathyroid hormone (PTH) therapy induced allograft integration while modulating angiogenesis at the allograft proximity. Here, we hypothesize that PTH-induced vascular modulation and the osteogenic effect of PTH are both dependent on endothelial PTH receptor-1 (PTHR1) signaling. To evaluate our hypothesis, we used multiple transgenic mouse lines, and their wild-type counterparts as a control. In addition to endothelial-specific PTHR1 knock-out mice, we used mice in which PTHR1 was engineered to be constitutively active in collagen-1α+ osteoblasts, to assess the effect of PTH signaling activation exclusively in osteoprogenitors. To characterize resident cell recruitment and osteogenic activity, mice in which the Luciferase reporter gene is expressed under the Osteocalcin promoter (Oc-Luc) were used. Mice were implanted with calvarial allografts and treated with either PTH or PBS. A micro-computed tomography-based structural analysis indicated that the induction of bone formation by PTH, as observed in wild-type animals, was not maintained when PTHR1 was removed from endothelial cells. Furthermore, the induction of PTH signaling exclusively in osteoblasts resulted in significantly less bone formation compared to systemic PTH treatment, and significantly less osteogenic activity was measured by bioluminescence imaging of the Oc-Luc mice. Deletion of the endothelial PTHR1 significantly decreased the PTH-induced formation of narrow blood vessels, formerly demonstrated in wild-type mice. However, the exclusive activation of PTH signaling in osteoblasts was sufficient to re-establish the observed PTH effect. Collectively, our results show that endothelial PTHR1 signaling plays a key role in PTH-induced osteogenesis and has implications in angiogenesis.

## 1. Introduction

The human body maintains a remarkable capacity for self-regeneration, particularly with respect to bone fractures. Nonetheless, 5% of all fractures fail to heal properly over the expected time course [1], thus resulting in what is commonly referred to as a nonunion fracture. Fractures occur at a rate of 116.3 per 10,000 patient-years for males and females older than 50 [2], which translates to approximately 7,500,000 annual occurrences of nonunion fractures across the developed world. Specifically, nonunion fractures are defined as the failure to heal, to achieve union within 9 months, or within 6 months if there are no radiological signs of healing, and thus further intervention will be required [3]. It is said that every nonunion fracture is unique, and there are several reasons why a fracture may not properly heal, including bacterial infection at the fracture site [4]. Four classifications of nonunions have been established based on the presentation of the surrounding tissue. Hypertrophic nonunions occur when there is still a biological capacity for union, due to viable bone ends and adequate vascularization, but there is a failure to heal due to mechanical instability [3]. Atrophic nonunion occurs when the vascularization at the bone ends has been altered, and typically there is no evidence of callus formation upon X-ray [5]. This presentation typically indicates a problem with the body’s regenerative processes. Pseudarthrosis can be characterized as atrophic or hypertrophic but will typically lead to the formation of a false joint over time [6]. Oligotrophic nonunions exhibit a combination of the characteristics of the three previous nonunion types. Studies have shown that in cases of atrophic nonunion, BMPs are expressed yet the cellular environment is disrupted [7], and fibrogenic collagen III is secreted to provide structure and stability in response to a lack of adequate bone regeneration [8]. The hallmark of atrophic nonunion is the formation of scar tissue [9,10], or fibrosis, which may be referred to as “scarful healing”. The goal of a successful therapeutic strategy to treat atrophic nonunion fractures must be to prevent or divert the scarful healing to scarless healing.

Bone regeneration in the craniofacial complex presents a notable challenge. Long bones have lower rates of nonunion and develop and self-repair by recapitulating endochondral ossification, where ossification builds upon a cartilaginous model. The cranial bone, however, develops via a unique process called ‘intramembranous ossification’, in which mesenchymal stem cells (MSCs) condense between two membranes that will later define the shape of the flat bone, and then differentiate directly to osteoblasts, which secrete bone matrix [11]. This particular process of development is also the way in which cranial bone fractures will heal; however, as a result of the sparse stem cell populations present in the thin marrow, spontaneous healing of cranial defects does not occur in humans, except in children under the age of two [12]. Similar challenges are encountered when treating cases of extensive mandibular bone loss, such as when the bone has been resected to remove malignancies, which can impair a patient’s ability to breath, eat, talk, and drink.

The current gold standard for treating nonunion factures is the implantation of self-bone tissue, or autografts. This approach ensures that the necessary regenerative elements are available and active to create scarless healing. Nevertheless, members of the orthopedic community have raised concerns about the use of autografts [13]; the obvious problem is the trauma caused to the donor site by the harvesting process, which can lead to chronic pain, sensory disturbance and even fracture of the donor site, and the added potential for common surgical complications, including infection and hypertrophic scarring [14]. Autografts may also be resorbed, especially in young patients [15], and on top of that, there is a limited amount of bone that can be harvested from a single patient. In addition, many patients with a nonunion fracture already suffer from a metabolic bone disease, including osteoporosis and Paget’s disease of the bones, thus making these patients poor donors.

A bone graft harvested from an allogeneic donor, referred to as an allograft, can be completely cleaned of the donor’s cellular components, while still retaining the necessary structural and biomechanical properties [16]. The wide range of potential allografts enables clinical solutions to be tailored to a specific patient by using computed tomography (CT), among other tools. Allografts lay the foundation for osteogenic regeneration, but they do not induce it, namely, they are ‘osteoconductive’, not ‘osteoinductive’. Thus, allograft implantation may lead to scarful healing. An advantage to using an allograft is that it is not recognized as non-self by the immune system, due to the complete removal of the donor’s cellular components, and as a result, there is no need for immunosuppression [17].

To promote the regeneration of bone when using an allograft, previous studies, including our own, have suggested that enhancement by parathyroid hormone (PTH) is beneficial for the induction of osteogenesis [18,19,20,21,22,23,24,25,26,27,28]. PTH is secreted by the parathyroid glands and functions as a regulator of calcium blood levels—when calcium levels drop, physiologic PTH secretion will induce bone resorption, which releases calcium into the blood from the body’s main calcium store: bone. Remarkably, intermittent administration of PTH induces the opposite response, an anabolic reaction leading to bone formation. As a result, a derivative of PTH, teriparatide, was approved for clinical use as a fracture preventative in osteoporotic patients at the beginning of the 21st century [18]. Since then it has been reported that PTH accelerates fracture healing in various animal models [19], as well as in humans [29]. The osteogenic effect of PTH is mediated by its receptor PTHR1, a G protein-coupled receptor that activates cyclic-AMP/protein kinase A (PKA) and is coupled with the protein kinase C (PKC) axis [20], which is also expressed in endothelial cells [30]. Intermittent PTH administration promotes MSC osteoblastogenesis directly, as evidenced by gene expression and reductions in osteoblast apoptosis. In addition, evidence indicates that the effect of PTH is mediated by the BMP/Smad1 pathway, and that this hormone is involved in the modulation of angiogenesis [21].

In our previous studies, utilizing mice bearing a critical size calvarial defect and implanted with allografts, we found that PTH therapy promotes the migration of the resident stem cell population to the graft area and induces differentiation, resulting in significantly more bone regeneration than in untreated mice [22]. This phenomenon has also been observed in studies on allografts in long bones [23,24]. In addition to the osteo-anabolic effect, we found that PTH therapy also modulates the evolving vascular tree that feeds the allograft. Studies have shown that PTH-enhanced structural allograft healing is associated with decreased mast cell mobilization and accumulation in the allograft region, and has resulted in a decrease in the extent of fibrosis that subsequently encapsulates the graft [25,31]. The mechanisms underlying the effect of PTH on osteogenesis have been investigated extensively, yet the role of PTH signal transduction in endothelial cells remains unclear. In this study, we hypothesize that the PTH-induced vascular modulation and the PTH osteogenic effect are dependent on endothelial PTHR1 signaling. To evaluate our hypothesis, we utilized three different transgenic mice lines: (1) mice in which the PTHR1 was knocked out of the endothelial cells using the VE-cadherin promoter, which is widely associated with hemogenic endothelium [32], (2) mice induced to constitutively express active PTHR1 exclusively in osteoblasts and (3) mice expressing the Luciferase reporter gene regulated by the Osteocalcin promoter (Oc-Luc) to allow for the monitoring of osteoblastic cell recruitment and differentiation (Figure 1) [22].

## 2. Materials and Methods

All experiments were performed according to the ARRIVE guidelines and were approved by The Hebrew University of Jerusalem Institutional Animal Care and Use Committee (IACUC #MD-14-13960). The mice were housed in groups of up to five mice per cage, had a 12 h light/12 h dark cycle and received water and standard mice food ad-libitum. Endothelial-specific PTHR1 knock-out mice (hereafter referred to as VEcadCRE/PTHR mice) were generated by crossing VE-cadherin-CRE (VEcadCRE) mice with PTHR1fx/fx mice, both of which have a C57BL/6 (C57) background. To examine PTHR activation exclusively in osteoblasts, we used mice expressing PTHR1 under the α1(I) collagen promoter, hereafter referred to as Col1ca-PTHR mice [26]. Genotyping was conducted on lysed ear punch samples by PCR using the following primers: 5′-CCTCCCCAGCCCCCACATGG-3′ and 5′-ATGAGGTCTGAGGTACATGGCTCTGA-3′ for the VEcadCRE mice, 5′-ATGAGGTCTGAGGTACATGGCTCTGA-3′ and 5′-ACATGGCCATGCCTGGGTCTGAGA-3′ for the PTHR1fx/fx mice, and 5′-TTCCACCACTGCCTCCCATTAT-3′ and 5′-CACCTGCCCTGTACAGGAAGAG-3′ for the Col1ca-PTHR mice. Transgenic mice with an FVB/N background, expressing the Luciferase gene, regulated by the Osteocalcin promoter (Oc-Luc), where crossed with Col1ca-PTHR mice and used for bioluminescence imaging. The luminescence of each mouse was verified by in vitro imaging of homogenized tail vertebra prior to in vivo imaging.

The calvarial defect model has been thoroughly described in previous publications [15,18,26], including by publication of an instructional video [33]. Briefly, circular allografts, 4.5 mm in diameter, were harvested from a mouse strain different from the host—C57BL/6 allografts were used in FVB/N hosts, and vice versa. Soft tissue was scraped off, and the grafts were decellularized by 70% ethanol, following extensive PBS washes. The allografts were stored at −70 °C at least 1 week prior to implantation. To create the calvarial model, 8-week-old female mice of each aforementioned experimental group were anesthetized using ketamine/dexmedetomidine (75 mg/0.5 mg per kg body weight), shaved and disinfected. A 5 mm full-thickness circular defect was induced, using the lambdoid suture as an anatomical reference. This defect size was chosen because it is the most commonly used in a mouse model [34,35,36]. To mimic the clinical setting, the allograft was placed in the defect without direct contact with the host bone and secured in place using fibrin gel (Tisseel Kit, Baxter AG, Vienna, Austria). The skin was then sutured. The animals were given preoperative (buprenorphine 0.1 mg/kg) and postoperative (carprofen 5 mg/kg) analgesic medications. Mice randomly assigned to the PTH-treated group were given 40 µg/kg body weight/day of teriparatide, for 21 consecutive days, as previously described [22,25], while mice assigned to the control group were given an equivalent PBS mock-treatment for the 3-week duration of the study.

The structural analysis of blood vessels was also presented in the video-based protocol mentioned previously [33]. One week after surgery, the mice were anesthetized by intraperitoneal administration of ketamine/dexmedetomidine, as stated above. The thoracic cavity was opened to expose the heart, and a needle was inserted into the left ventricle. The vasculature was flushed with 0.9% heparinized saline (100 U/mL), followed by 10% neutral buffered formalin. Then, a radio-opaque silicone contrast agent was infused—Microfil MV-122 (Flow-tech Inc., Carver, MA, USA). Samples were stored at 4 °C overnight to allow curing of the silicone-based contrast agent. The calvarial region was dissected and then decalcified, to enable imaging of the vasculature only, using ethylenediaminetetraacetic acid (EDTA—Sigma-Aldrich, St Louis, MO, USA) for a minimum of 4 weeks. Samples were scanned using a desktop µCT 40 (Scanco Medical AG, Bassersdorf, Switzerland), at an X-ray energy of 55 kVp, intensity of 145 µA, 1000 projections per 360°, integration time of 200 ms and voxel size of 12 µM. Two-dimensional slices were reconstructed and used to locate the anatomical site of the allograft. The contrast agent was segmented from soft tissue using a global threshold procedure, followed by application of a Gaussian filter (sigma [σ] = 2.0 and support = 2) to suppress noise in the volumes. Then a thickness map was generated using standard IPL software and the “dt_object_param” function. Statistical analysis was performed by 2-way ANOVA with Holm–Šídák’s multiple-comparisons test, following the removal of outliers defined as values exceeding the range of average ± 2 ∗ SE.

Analysis of bone formation was performed 4 weeks post-allograft implantation, a time point chosen based on our previous research in a mouse calvarial defect model [22]. Mice were euthanized and the calvarial region was isolated. Samples were scanned and analyzed in accordance to accepted guidelines [37], using a previously published method [38]. For the µCT analysis of bone formation, the following parameters were used: energy of 70 kVp, intensity of 114 µA, 1000 projections per 360°, integration time of 200 ms and voxel size of 20 µM. Again, the bone tissue was segmented from the soft tissues and a Gaussian filter was applied, this time using sigma [σ] = 0.8 and support = 1. Samples were rotated to a standard position and a volume of interest was defined in reference to the calvarial defect. The bone mineral density (BMD) was measured by comparing the scanned samples to phantoms containing a known density of hydroxyapatite [39]. Statistical analysis was performed using an unpaired *t*-test for comparison between two experimental groups.

To assess osteogenic activity, bioluminescence imaging (BLI) was performed using transgenic mice expressing the Luciferase gene regulated by the osteocalcin promoter. As a result, the BLI signal represents: (a) resident stem cell migration to the cell-free allograft, and (b) stem cell differentiation towards the osteogenic lineage [15,26,30]. Imaging was performed on day 4, 7, 10, 14, 21 and 28 post-implantation of the calvarial allografts, for both FVB/N mice systemically treated with PTH, in which case all cells were exposed to PTH, and for Col1ca-PTHR animals given PBS, in which PTHR1 signaling was constitutively active exclusively in osteogenic cells. For bioluminescence imaging, mice were anesthetized by 1–3% isoflurane mixed with 100% medical grade oxygen, and then received an intraperitoneal injection of Beetle luciferin (Perkin-Elmer, Waltham, MA, USA), at a dose of 126 mg/kg of body weight. Imaging was delayed for 5 min to allow sufficient time for the intended chemical reaction to occur, the luciferin oxidation. The light emission was quantified using an IVIS-spectrum system (Perkin-Elmer, Waltham, MA, USA), using the automatically-set exposure time. As mice can vary considerably in their transgene expression, we used the tail vertebrae as an internal control for each mouse, a method that has also been previously reported [22].

## 3. Results

### 3.1. The PTH Osteogenic Effect Is Diminished When PTHR1 Is Deleted in Endothelial Cells

In a comparison of wild-type C57BL/6 (C57) mice treated with PTH and PBS, at four weeks post-allograft implantation, significantly higher bone volume was measured in mice given PTH (Figure 2A,B). In VEcadCRE/PTHR mice, who share the C57BL/6 background but lack the PTHR1 in endothelial cells, administration of PTH therapy yielded no significant difference in bone volume formation compared to PBS administration. The bone connectivity and density of C57 wild-type mice was not significantly higher in the PTH-treated group, although the mean of the PTH-treated group is higher (Figure 2C). This trend was not seen in PTH-treated VEcadCRE/PTHR mice, as they had a lower mean bone connectivity and density, compared to mice treated with PBS, although this difference was also not statistically significant. The bone mineral density (BMD) was found to be similar across all studied groups regardless of the treatment received (Figure 2D).

### 3.2. Induction of the PTH Axis in Osteogenic Cells Alone Is Not Sufficient to Induce Bone Formation

Next, we sought to investigate whether activation of the PTHR exclusively in osteoprogenitors, and therefore not in endothelial cells, would be sufficient to induce bone regeneration. Col1ca-PTHR mice, with PTHR signal transduction constitutively activated solely in collagen-1α+ osteoprogenitors, were compared to the FVB/N wild-type mice that were given daily systemic PTH (Figure 3A). As such, in the FVB/N wild-type mice treated with PTH, the resident MSCs, osteoprogenitors, endothelial cells, hematopoietic cells and others were all exposed to the PTH stimulus. It is important to note that the constitutive activation of PTHR in osteoprogenitors does not fully mimic the clinical use of teriparatide, differing from continuous PTH infusion in two aspects: (1) only the collagen-1α+ osteoprogenitor cells have a constantly active PTH pathway, without any other cell types impacted by this activation, and (2) we cannot directly measure the extent of activation or correlate this with a known administered dosage of PTH. Four weeks after the implantation of the allograft, significantly higher bone volume was observed in the FVB/N wild-type mice given PTH (FVBn + PTH), compared to Col1ca-PTHR mice (Figure 3B). There was also a significantly higher mean bone connectivity and density value in the FVBn + PTH group (Figure 3C), which suggests a higher rate of graft connectivity to the host bone. As seen previously with the C57 wild-type and VEcadCRE/PTHR mice, the mean bone mineral density (BMD) was similar across both groups (Figure 3D).

To further characterize the difference between systemic PTH administration and the targeted activation of the PTHR in osteoprogenitors, we crossed Col1ca-PTHR mice with FVB/N mice harboring the Luciferase reporter gene under the Osteocalcin promotor (Col1ca-PTHR/Oc-Luc). The FVB/N mice harboring the Luciferase reporter gene under the Osteocalcin promotor (Oc-Luc) were administered PTH for comparison (Oc-Luc + PTH). Based on the data collected from bioluminescence imaging, osteogenic activity at the allograft site is significantly higher in Col1ca-PTHR mice at 4, 7, 10 and 21 days post-surgery, compared to their wild-type counterparts (Figure 4A,B). However, we also found that systemic osteogenic activity was significantly greater in Col1ca-PTHR mice, and thus potentially independent of bone fracture repair or allograft-mediated cranial healing. In order to appropriately compare the two groups, the highest signal in each Col1ca-PTHR tail vertebrae was used to internally normalize the cranial activity of each animal. Once normalized, the previously observed relationship was no longer present, and it was found that osteogenic activity on day 3 and day 7 post-implantation was significantly greater in the Oc-Luc + PTH group compared to Col1ca-PTHR mice (Figure 4C). Following this increase, there was a return to similar intensities by day 28.

### 3.3. The PTH Modulation of the Vascular Tree Is Reliant on Endothelial PTHR Signaling

To assess the effect of PTH on the evolving vascular tree feeding the cranial allograft, a µCT-based structural analysis was utilized (Figure 5A). A higher number of small-diameter blood vessels was observed in C57BL/6 wild type mice given PTH (C57 + PTH), compared to the VEcadCRE/PTHR mice, in which the PTHR was deleted in VE-cadherin+ endothelial cells, also given PTH. Specifically, the C57 + PTH group had a significantly greater amount of blood vessels with diameters of 48 µm, 60 µm, 72 µm, and 96 µm. On the other hand, large blood vessels were significantly more abundant in the VEcadCRE/PTHR + PTH group, specifically blood vessels with diameters of 144 µm and greater than 156 µm (Figure 5B).

Remarkably, Col1ca-PTHR mice, in which targeted activation of PTHR occurred in osteoprogenitors only, showed a similar pattern as the wild-type C57BL/6 mice treated with PTH. There were frequently more small-diameter blood vessels when comparing the Col1ca-PTHR group to VEcadCRE/PTHR mice given PTH. There were significantly more blood vessels with a diameter of 60 µm and 84 µm in the Col1ca-PTHR group. In addition, compared to the VEcadCRE/PTHR mice, Col1ca-PTHR mice had significantly less large blood vessels with a diameter of 168 µm, 180 µm, 216 µm, and 228 µm. This indicates that the targeted activation of PTHR in osteoprogenitors results in a similar angiogenic profile as wild-type animals receiving systemic PTH.

## 4. Discussion

PTH has been well-established as an inducer of osteogenesis in the osteoporotic patient and has been used in the treatment of various bone fractures in both osteoporotic and non-osteoporotic patients. This study, as well as the work of others, demonstrates that PTH also plays a role in directing the vascular tree towards intact bone [40] and towards the site of a fracture [21,25]. The osteogenic process required to achieve bone formation has been shown to rely on active vascularization, which is modulated by PTH both directly [41] and indirectly [31]. Particularly in allograft-based fracture healing, we have shown that the modulation of the vasculature formed under PTH treatment is associated with a lessened accumulation of mast cells, which in turn results in decreased fibrogenesis [42]. In a previous publication, we demonstrated that intermittent systemic PTH administration in a mouse model, following the implantation of an allograft in the calvaria, induced the formation of narrower blood vessels, compared to mice that did not receive PTH treatment; this led to less scar tissue formation and superior host bone formation around the defect site in the PTH-treated group [25]. Still, the cellular mechanisms by which PTH drives the modulation of angiogenesis remains unclear.

The expression of PTHR in endothelial cells was established about 20 years ago [43] and its protein kinase A (PKA) and protein kinase C (PKC)-mediated functions have been detailed thoroughly [44], as has the hormone’s dose-dependent effect on endothelial cells [45]. Since then, researchers have demonstrated the effect of PTH in different disease models, including uremic vascular changes [46], yet we are the first to demonstrate the effect on PTH on endothelial cells in the context of bone regeneration. In this study, we used a cranial allograft model to characterize the role that endothelial PTHR plays in the osteogenic effect of PTH, as well as the effect of this receptor on angiogenesis. We utilized transgenic mice to show that when PTH is deleted in endothelial cells, the hormone no longer induces the superior bone formation found in previous studies, as demonstrated by µCT (Figure 2). When PTHR signaling activation is targeted exclusively in osteoprogenitors, and therefore not in endothelial cells, again, PTH-driven osteogenesis was not observed (Figure 3).

To further understand the effect of PTH on osteogenesis we used BLI. These results indicated that the activation of resident stem cells seen in wild-type animals given PTH is not comparable to osteoprogenitor PTHR activation (Figure 4). The heightened systemic expression seen in Col1ca-PTHR mice may lead to the speculation that the osteogenic PTH axis had been turned on to the maximum rate for bone remodeling [28], but the authors feel that this assertion still requires further investigation. The visible difference in the tail signal between the wild-type animals and the Col1ca-PTHR mice does suggest that there may be other factors besides the introduction of the calvarial defect, which resulted in greater expression. To this point, in a 2001 JCI publication presenting the Col1ca-PTHR strain, the main finding was that compared with wild-type mice, the Col1ca-PTHR strain exhibited an increase in trabecular bone formation and a decrease in cortical bone formation [26]. This study was in the context of intact bone, not of a bone defect, but when the researchers specifically examined the calvarial intermembranous bone, they reported an unchanged calvaria thickness compared to the wild-type control. They also found that the Col1ca-PTHR mice had more porous bone at sites where cortical bone had been replaced by woven bone and noted significant increases in bone formation on the endosteal surface and decreases in bone formation on the periosteal surface. Based on the context provided by Calvi et al., even though that when normalized to the signal of the tail vertebrae, there was a significantly lower signal at day 3 and day 7 post-implantation in the Col1ca-PTHR mice, we believe that the constitutive activation of PTHR in osteoprogenitors does not result in bone resorption but it increases bone metabolism in a way that is unique to what is seen in the wild-type mice.

Lastly, the results from the vascular µCT structural analysis highlights that PTH therapy, which has been shown to induce the formation of narrower blood vessels when given to wild-type mice, has a lessened effect when the PTHR has been deleted from endothelial cells (Figure 5). Interestingly, we did note that the targeted activation of PTHR signaling in osteoprogenitors is sufficient to induce vascular modulation similar to what has been observed in wild-type mice. It should be clarified that these experiments were conducted using mice of a different genetic background: C57BL/6 for the VEcadCRE/PTHR mice and FVB/N for the Col1ca-PTHR mice. Nevertheless, this result suggests that osteoprogenitors have a role in directing angiogenesis, such as they do in the postnatal growth plate [47]. The effect that PTH has on angiogenesis may involve additional pathways; a previous study proposed that PTH induces oxidative stress in endothelial cells [48], yet the present study has shown that its effect is dependent on endothelial PTHR expression. However, further research is needed to continue to understand the role of PTHR in endothelial growth and function, especially in the context of osteo-endothelial pathways.

A limitation of our experimental design is the fact that there is a difference between the constitutive activation of PTHR signaling and the intermittent administration of PTH used in the clinical setting. Constitutive PTHR activation has been found to have a different effect on osteogenesis [27], when compared to intermittent PTH administration, and at this point the differing effect on angiogenesis has not been fully characterized [26]. In addition, we are unable to directly measure the extent to which PTHR signaling has been activated, which inhibits our ability to estimate a comparable “dose” of PTH. These limitations present challenges in translating this data to the clinical setting, but we believe that this study provides additional support for the continued research into the use of PTH for allograft-based fracture healing.

## 5. Conclusions

The purpose of our study was to determine the role of endothelial cell PTHR in PTH-induced osteogenesis and to better characterize the role of this receptor in the ability of PTH to modulate angiogenesis. For this purpose, we used a murine model of cranial defect regeneration by allograft implantation. We found that the deletion of endothelial PTHR negates the PTH osteogenic effect and that the targeted activation of the PTH axis in osteoprogenitors is incapable of restoring this effect. In addition, our results demonstrate that upon deleting the endothelial PTHR, the previously demonstrated PTH-induced formation of narrower blood vessels feeding the allograft proximity is impeded. However, in the case related to angiogenesis, the targeted activation of PTHR exclusively in osteoprogenitors results in similarly narrow blood vessels to the control. We concluded that the PTH effect on bone formation is, at least partially, mediated by endothelial PTHR signaling. Additionally, we found that the modulation of blood vessels feeding the allograft proximity is dependent on the presence of endothelial PTHR, though osteoblastic PTHR signaling also appears to play a role in shaping the evolving vascular tree. The effect of PTH on angiogenesis in the context of bone regeneration has previously been characterized, but we are the first to delve into the specific cellular mechanisms of signaling. Next steps include using a clinically relevant sized defect for better translation to the clinical setting and further investigation of the specific biological processes that allow PTH to induce osteogenesis and modulate angiogenesis, in the hope that this can be further optimized. We believe that our research will work in conjunction with both past and future studies to help overcome the current clinical challenges relating to PTH-induced bone regeneration.

## Figures and Tables

**Figure 1 cells-11-00897-f001:**
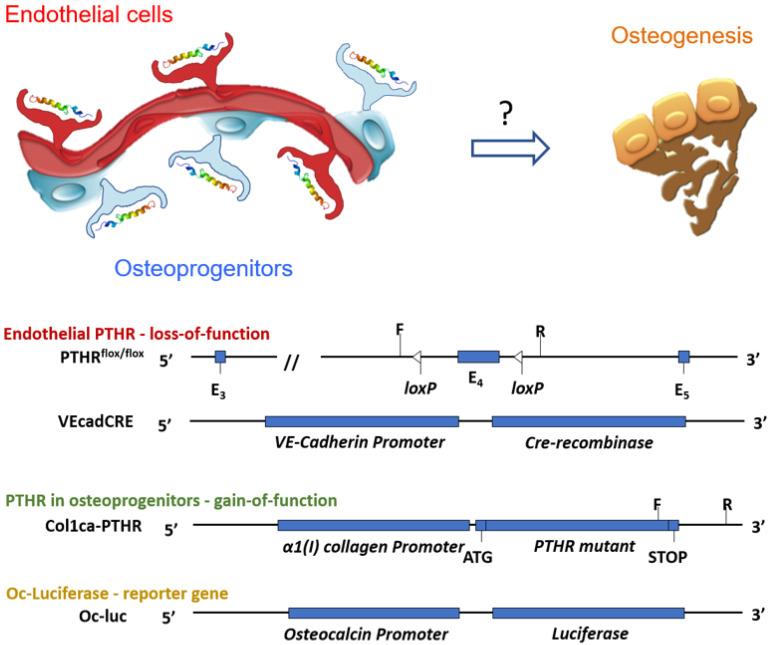
Experimental overview. The aim of this study was to further elucidate the mechanism of the PTH effect on osteogenesis and angiogenesis, specifically in the context of allograft-mediated cranial defect healing. We hypothesize that the PTHR1 in endothelial cells plays a critical role in PTH-induced bone formation, as well as in the hormone’s ability to modulate the vascular tree feeding the graft. To evaluate this hypothesis, we used several lines of transgenic mice: (1) mice in which a CRE-flox system was implemented to delete the PTHR1 in endothelial VE-cadherin+ cells, (2) mice that exclusively over-express constitutively active PTHR1 in collagen-1α+ osteoprogenitors, and (3) mice expressing the Luciferase reporter gene in osteocalcin+ osteoprogenitor cells.

**Figure 2 cells-11-00897-f002:**
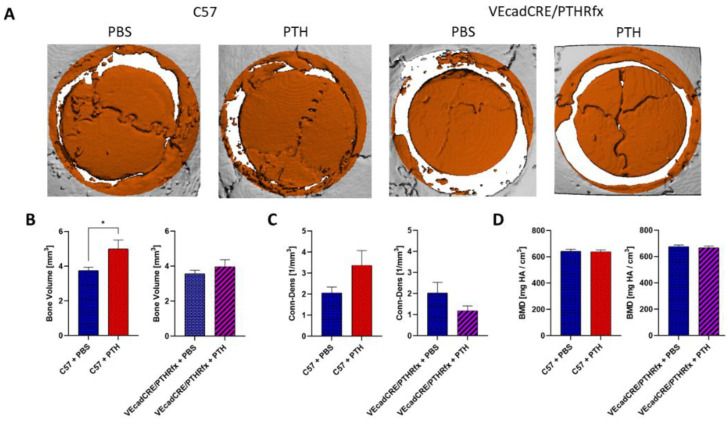
The deletion of PTHR in endothelial cells diminished the hormone’s osteogenic effect. Wild-type C57BL/6 mice (C57) and C57BL/6 mice engineered with CRE-recombinase under the endothelial VE-cadherin promoter and floxed PTH-Receptor 1 gene (VEcad/PTHRfx) underwent a circular 5 mm defect in the skull, followed by bone allograft implantation. Four weeks later, and one week after a 3-week treatment with either PTH or PBS (control group), significantly greater bone volume was observed in the wild-type C57 mice that received PTH, but no difference in bone volume was observed between the transgenic mice treatment groups (**A**,**B**), *n* = 6, * *p*-value < 0.05, *t*-test. Bone connectivity and density was higher in wild-type C57 mice given PTH, compared to the counterpart control group (**C**), *n* = 6, *p*-value = 0.057, *t*-test. Bone mineral density (BMD) was similar across all groups (**D**).

**Figure 3 cells-11-00897-f003:**
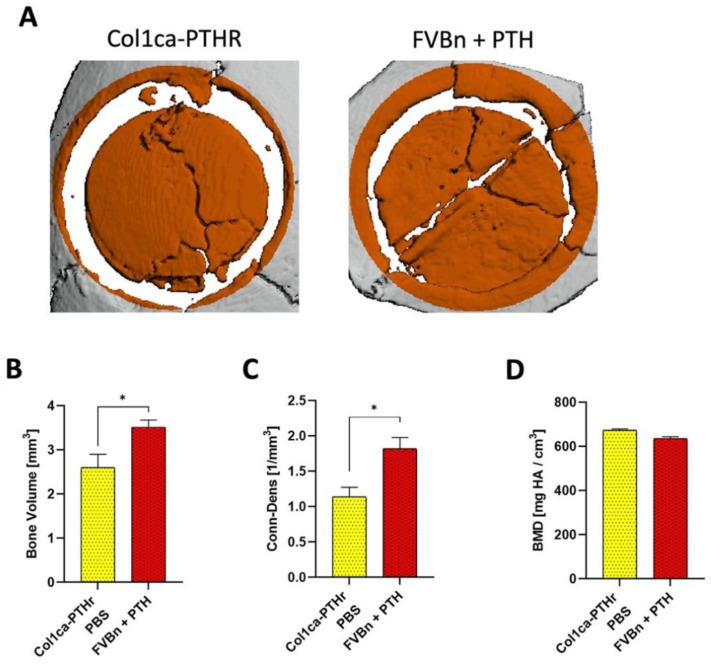
PTHR signaling activated in osteoprogenitors induced osteogenesis to a lesser degree compared to systemic PTH administration. Bone allografts were implanted in wild-type FVB/N mice given PTH (FVBn + PTH) and in FVB/N mice engineered with constitutively active intramembranous portions of the PTHR under collagen-1α (Col1ca-PTHR). Mice were euthanized four weeks later, and bone volume was significantly higher in the wild-type mice treated with systemic PTH, compared to Col1ca-PTHR mice (**A**,**B**), *n* = 6, * *p*-value < 0.05, *t*-test. Furthermore, bone connectivity and density were significantly higher in the FVBn + PTH group (**C**), *n* = 6, * *p*-value < 0.05, *t*-test. Bone mineral density (BMD) was similar in both groups (**D**).

**Figure 4 cells-11-00897-f004:**
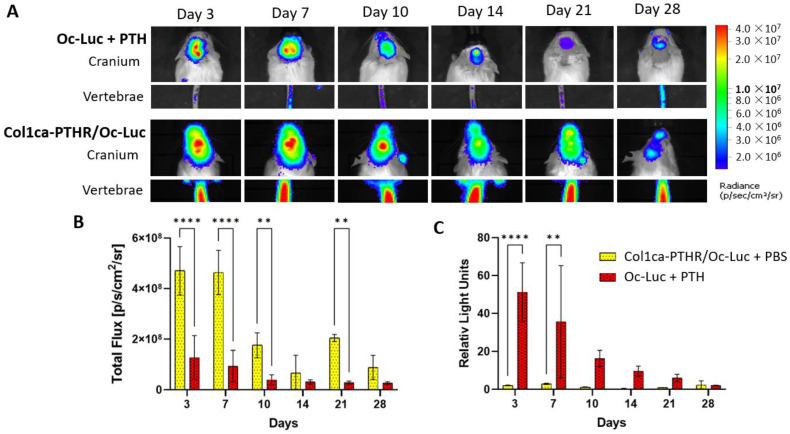
The induction of host cell differentiation is greater when PTH is administered systemically, compared to PTHR activation in osteoprogenitors alone. The Col1ca-PTHR mice were crossed with Osteocalcin-Luciferase (Oc-Luc) mice. Bioluminescence imaging was performed at designated time points following allograft implantation, using the beetle luciferin substrate for luminescence (**A**). Quantitative analysis was performed first in the cranial region (**B**), *n* = 7, ** *p*-value < 0.01, **** *p*-value < 0.001, two-way ANOVA, and then normalized to each animal’s tail vertebrae, which acted as a non-fractured anatomical site to normalize luminescence (**C**), *n* = 7, ** *p*-value < 0.01, **** *p*-value < 0.001, two-way ANOVA.

**Figure 5 cells-11-00897-f005:**
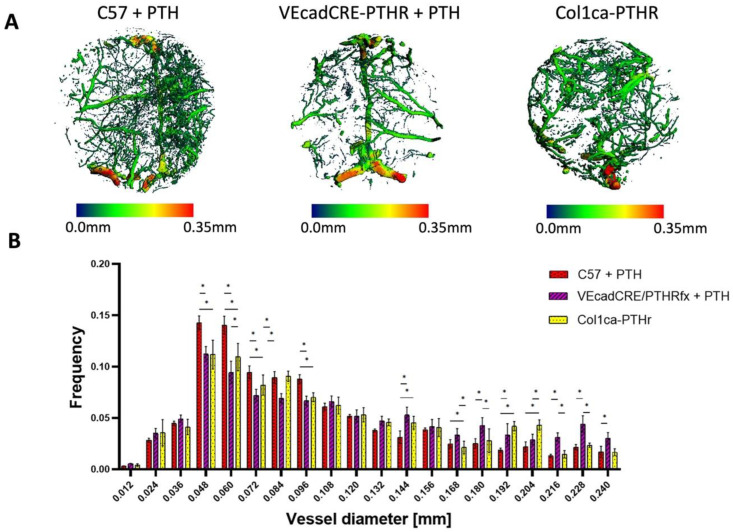
Systemic PTH administration increases the formation of small-diameter blood vessels, while PTHR deletion in endothelial cells attenuates this PTH-induced effect, but PTHR activation in osteoprogenitors alone is sufficient to re-establish this effect. One week following the calvarial graft implantation, mice were anesthetized, and a radio-opaque contrast agent was systemically administrated. After the rubbery contrast agent was set to cure, the calvarial region was harvested and decalcified, so the vasculature could be analyzed by µCT. First, a 3-dimensional thickness map was generated (**A**), and the frequency of each vessel-diameter was calculated (**B**), *n* = 8, * *p*-value < 0.05, two-way ANOVA.

## Data Availability

The data presented in this study are available upon request from the corresponding author.

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
