# Peer review of "PTH-Induced Bone Regeneration and Vascular Modulation Are Both Dependent on Endothelial Signaling"

_cells, 2022, doi:10.3390/cells11050897_

Round 1

Reviewer 1 Report

Well written. Original. Important for the field.

Author Response

Comments from Reviewer 1

  • Well written. Original. Important for the field.

Response: We thank the reviewer for their positive comments and appreciate them taking the time to review our manuscript.

Reviewer 2 Report

I have examined the manuscript entitled 'PTH-induced bone regeneration and vascular modulation are both dependent on endothelial signaling' . This manuscript provides information about the Effect of PTH on scarless bone defect regeneration via endothelial signaling. However, revisions are necessary before it should be published.

Comments to the Author

  1. Please describe the method followed for the calculations of bone density and what are the parameters used for CT technique.
  2. English needs revision. 

    The topic of the manuscript might be interesting to the readers of this journal and it falls within its scope. Therefore, I suggest that the manuscript may be accepted for publication after major revision.

    Thanks!

Author Response

Comments from Reviewer 2

  • I have examined the manuscript entitled ‘PTH-induced bone regeneration and vascular modulation are both dependent on endothelial signaling’. This manuscript provides information about the effect of PTH on scarless bone defect regeneration via endothelial signaling. However, revisions are necessary before it should be published.

Response: We appreciate the reviewer taking the time to review our manuscript and hope that our revisions have adequately addressed the reviewer’s comments.

  • Please describe the method followed for the calculations of bone density and what are the parameters used for CT technique.

Response: We greatly appreciate this comment as the parameters used for the bone formation µCT analysis were omitted unintentionally. The bone mineral density is a value calculated by the µCT analysis program Imaging Processing Language (IPL), comparing the scanned samples to phantoms containing a known density of hydroxyapatite. The µCT scanner was regularly calibrated using the phantoms per standard operating procedure. Based on this comment, we have added the following sentences to the Materials and Methods section and a citation to a paper which further explains the use of hydroxyapatite phantoms with the µCT 40 scanner used in this study should the reader want to understand the process more in depth.

Line 206-208: For the µCT analysis of bone formation, the following parameters were used: energy of 70 kVp, intensity of 114 µA, 1,000 projections per 360°, integration time of 200 ms and voxel size of 20 µM.

Line 211-212: The bone mineral density (BMD) was measured by comparison of the scanned samples to phantoms containing a known density of hydroxyapatite [39].

  1. Nazarian, A.; Snyder, B.D.; Zurakowski, D.; Muller, R. Quantitative micro-computed tomography: a non-invasive method to assess equivalent bone mineral density. Bone 2008, 43, 302-311, doi:10.1016/j.bone.2008.04.009.

  • English needs revision.

Response: The manuscript was sent for revision by a professional scientific editor.

  • The topic of the manuscript might be interesting to the readers of this journal, and it falls within its scope. Therefore, I suggest that the manuscript may be accepted for publication after major revision.

Response: We greatly appreciate the opportunity to revise our manuscript and appreciate this reviewer’s suggestions.  

Reviewer 3 Report

The manuscript explores some mechanistic aspects of the effect of PTH therapy in allograft integration. Using elegant mouse genetics models, the authors show that expression of the PTH receptor (PTHR) in vascular endothelial cells is necessary for PTH-dependent bone formation at the allograft site. Expression of a constitutively active PTH receptor in osteoblasts in transgenic mice resulted in less bone formation at the allograft site compared to treatment of wild-type animals with intermittent PTH. Concerning angiogenesis, specific inactivation of the PTH receptor in vascular endothelial cells reduced PTH-induced formation of narrow blood vessels, but this effect was not seen in transgenic mice expressing the constitutively active PTH receptor in osteoblasts. The authors conclude that the PTH osteoanabolic and angiogenic effects during allograft integration are, at least partially, mediated by endothelial PTH receptor signaling.

The manuscript is well written and addresses an important aspect of the osteoanabolic response to PTH treatment, i.e., the impact of treatment on angiogenesis. For the most part, the data support the authors’ conclusions. Some additional consideration of PTH effects in the interpretation of the results would strengthen the manuscript.

Specific comments:

  1. Expression of a constitutively active PTHR in osteoblasts is equivalent to infusion with PTH, which would be expected to stimulate bone resorption. Adding some data concerning this effect (expression of RANKL by gene expression monitoring or ELISA; circulating bone resorption markers) would be a valuable addition. At the very least, the authors should consider this effect in the interpretation of the data in Figure 2 and in the discussion.
  2. The bioluminescence data is overinterpreted. The authors mention that in transgenic mice expressing the constitutively active PTH receptor in osteoblasts, ‘… only the osteoprogenitors were induced…’. The resolution of the data generated by bioluminescence does not allow to discriminate between cell types mediating the effect. This should be toned down or rephrased in the discussion.
  3. minor: Figure 3C should be mentioned at the end of the sentence on line 229.

Author Response

Comments from Reviewer 3

  • The manuscript explores some mechanistic aspects of the effect of PTH therapy in allograft integration. Using elegant mouse genetics models, the authors show that expression of the PTH receptor (PTHR) in vascular endothelial cells is necessary for PTH-dependent bone formation at the allograft site. Expression of a constitutively active PTH receptor in osteoblasts in transgenic mice resulted in less bone formation at the allograft site compared to treatment of wild-type animals with intermittent PTH. Concerning angiogenesis, specific inactivation of the PTH receptor in vascular endothelial cells reduced PTH-induced formation of narrow blood vessels, but this effect was not seen in transgenic mice expressing the constitutively active PTH receptor in osteoblasts. The authors conclude that the PTH osteoanabolic and angiogenic effects during allograft integration are, at least partially, mediated by endothelial PTH receptor signaling. The manuscript is well written and addresses an important aspect of the osteoanabolic response to PTH treatment, i.e., the impact of treatment on angiogenesis. For the most part, the data support the authors’ conclusions. Some additional consideration of PTH effects in the interpretation of the results would strengthen the manuscript.

Response: We greatly appreciate this reviewer taking the time to provide such insightful comments and believe that the quality of the manuscript has benefitted greatly as a result.

  • Expression of a constitutively active PTHR in osteoblasts is equivalent to infusion with PTH, which would be expected to stimulate bone resorption. Adding some data concerning this effect (expression of RANKL by gene expression monitoring or ELISA; circulating bone resorption markers) would be a valuable addition. At the very least, the authors should consider this effect in the interpretation of the data in Figure 2 and in the discussion.

Response: That is a very interesting point, and the answer is not trivial. We greatly appreciate that this reviewer presented this comment and we have added to several sections in the manuscript to address this point.  

Line 251-256: It is important to note that the constitutive activation of PTHR in osteoprogenitors does not fully mimic the clinical use of teriparatide, differing from continuous PTH infusion in two aspects: 1) only the collagen-1α+ osteoprogenitor cells have a constantly active PTH pathway, without any other cell types impacted by this activation, and 2) we cannot directly measure the extent of activation or correlate this with a known administered dosage of PTH.

Line 363-382: The heightened systemic expression seen in Col1ca-PTHR mice may lead to the speculation that the osteogenic PTH axis had been turned on to the maximum rate on bone re-modeling [28], but the authors feel that at this time this assertion requires further investigation. The visible difference in the tail signal between the wild-type animals and the Col1ca-PTHR mice does suggest that there may be other factors besides the introduction of the calvarial defect which have resulted in greater expression. To this point, in a 2001 JCI publication presenting the Col1ca-PTHR strain, the main finding was that, compared with wild-type mice, the Col1ca-PTHR strain exhibited an increase in trabecular bone formation and a decrease in cortical bone formation [26]. This study was in the context of intact bone, not of a bone defect but when the researchers specifically examined the calvarial intermembranous bone, they reported an unchanged calvaria thickness compared to the wild-type control. They also found that the Col1ca-PTHR mice had more porous bone at sites where cortical bone had been replaced by woven bone and noted significant increases in bone formation in the endosteal surface and decreases in bone formation in the periosteal surface. Based on the context provided by Calvi et al., even though that when normalized to signal of the tail vertebrae there was a significantly lower signal at day 3 and day 7 post-implantation in the Col1ca-PTHR mice, we believe that the constitutive activation of PTHR in osteoprogenitors is not resulting in bone resorption but is in-creasing bone metabolism in a way that is unique to what is seen in the wild-type mice.

Line 398-407: A limitation of our experimental design is the fact that there is a difference between the constitutive activation of PTHR signaling and the intermittent administration of PTH used in the clinical setting. Constitutive PTHR activation has been found to have a different effect on osteogenesis [27], when compared to intermittent PTH administration, and at this point the differing effect on angiogenesis has not been fully characterized [26]. In addition, we are unable to directly measure the extent of which PTHR signaling has been activated, which inhibits our ability to estimate a comparable “dose” of PTH. These limitations present challenges to translating this data to the clinical setting, but we believe that this study provides additional support for the continued research into the use of PTH for allograft-based fracture healing.

  1. Calvi, L.M.; Sims, N.A.; Hunzelman, J.L.; Knight, M.C.; Giovannetti, A.; Saxton, J.M.; Kronenberg, H.M.; Baron, R.; Schipani, E. Activated parathyroid hormone/parathyroid hormone-related protein receptor in osteoblastic cells differentially affects cortical and trabecular bone. J Clin Invest 2001, 107, 277-286, doi:10.1172/JCI11296.
  2. Roche, B.; Vanden-Bossche, A.; Malaval, L.; Normand, M.; Jannot, M.; Chaux, R.; Vico, L.; Lafage-Proust, M.H. Parathyroid hormone 1-84 targets bone vascular structure and perfusion in mice: impacts of its administration regimen and of ovariectomy. J Bone Miner Res 2014, 29, 1608-1618, doi:10.1002/jbmr.2191.

  • The bioluminescence data is overinterpreted. The authors mention that in transgenic mice expressing the constitutively active PTH receptor in osteoblasts, ‘… only the osteoprogenitors were induced…’. The resolution of the data generated by bioluminescence does not allow to discriminate between cell types mediating the effect. This should be toned down or rephrased in the discussion.

Response: Indeed, the bioluminescence itself does not allow the resolution to determine which cell type emit the signal. We also rephrased several of the sentences where we stated the referenced phrase and replaced it with the following: “the targeted activation of PTHR in osteoprogenitors”. In addition, we revised out Discussion section and changed the language we used regarding the BLI data so that we were not overstating our results. We also added the following sentence to convey to the reader that while the BLI data is important, we caution against making too many conclusions solely based on it. We also believe that the added explanation in our Discussion section, based on the previous comment from this reviewer, provides a more contextual view of our data.

Line 363-366: The heightened systemic expression seen in Col1ca-PTHR mice may lead to the speculation that the osteogenic PTH axis had been turned on to the maximum rate on bone re-modeling [28], but the authors feel that at this time this assertion requires further investigation.

  1. Jilka, R.L.; O'Brien, C.A.; Bartell, S.M.; Weinstein, R.S.; Manolagas, S.C. Continuous elevation of PTH increases the number of osteoblasts via both osteoclast-dependent and -independent mechanisms. J Bone Miner Res 2010, 25, 2427-2437, doi:10.1002/jbmr.145.

  • Figure 3C should be mentioned at the end of the sentence on line 229.

Response: Indeed! Thank you for this comment. The reference to Figure 3C was added at the end of the sentence, now on Line 275 of the revised manuscript.

Reviewer 4 Report

The article entitled “PTH-induced bone regeneration and vascular modulation are both dependent on endothelial signaling” , this study they pothesized that the PTH-induced vascular modulation and the PTH osteogenic effect are dependent on endothelial PTHR1-signaling.

Below are some suggestions:

In the Abstract:

- The abstract is well written, but I suggest better describing the methodology mainly in relation to the experimental groups

In the Introduction:

- In general, the introduction is long. The authors could better explain the term "nonunion fracture" or replace it as it is not a commonly used terminology in the field of bone regeneration.

In the Materials and Methods:

- An illustrative image of the randomization of the groups could be made as well as the experimental design.

- Why was the 5mm calvaria defect performed and not the 8mm?

- Does the bone repair process in rats normally occur in seven weeks, why was the four weeks period used in the research? I suggest a justification.

In the Results:

- In figure 1 I suggest making two plates separating the images from the grafhics for better visibility and quality.

- I suggest increasing the size of the graphics of all figures as well as the quality to facilitate and improve the visualization

- Results can be better described, are confusing and very succinct

In the Discussion:

- The discussion could be written according to and in the same order as the data presented in the result, including the limitations of the study.

In the Conclusion:

- Change the conclusion into final considerations, including the purpose of the research, its main results, the conclusions and future clinical perspectives.

Author Response

Comments from Reviewer 4

  • The article entitled “PTH-induced bone regeneration and vascular modulation are both dependent on endothelial signaling”, this study they hypothesized that the PTH-induced vascular modulation and the PTH osteogenic effect are dependent on endothelial PTHR1-signaling.

Response: We thank the reviewer taking the time to review our manuscript.

  • Abstract: The abstract is well written, but I suggest better describing the methodology mainly in relation to the experimental groups.

Response: We appreciate this comment and have added the following details in the abstract for better clarity of the methodology.

Line 20-27: To evaluate our hypothesis we used multiple transgenic mouse lines, and their wild-type counterparts as a control. In addition to endothelial-specific PTHR1 knock-out mice, we used mice in which PTHR1 was engineered to be constitutively activate in collagen-1α+ osteoblasts, to assess the effect of PTH signaling activation exclusively in osteoprogenitors. To characterize resident cell recruitment and osteogenic activity, mice in which the Luciferase reporter gene is expressed under the Osteocalcin promoter (Oc-Luc) were used.

  • Introduction: In general, the introduction is long. The authors could better explain the term "nonunion fracture" or replace it as it is not a commonly used terminology in the field of bone regeneration.

Response: We appreciate this feedback. The term ‘nonunion’ is commonly used in publications presented in the field of bone tissue engineering and bone regeneration. To provide additional clarity on the definition of the term ‘nonunion’, and to address the different types of nonunion fractures, we have added the following explanation to the manuscript and the relevant references. We appreciate you presenting this suggestion and hope that the additional detail we have added to the manuscript is beneficial.   

Line 47-66: Specifically, nonunion fractures are defined as the failure to heal, to achieve union, within 9 months, or within 6 months if there are no radiological signs of healing and will thus require further intervention [3]. It is said that every nonunion fracture is unique, and there are several reasons why a fracture may not properly heal, including bacterial infection at the fracture site [4]. Four classifications of nonunions have been established based on the presentation of the surrounding tissue. Hypertrophic nonunions occur when there is still a biological capacity for union, due to viable bone ends and adequate vascularization, but there is a failure to heal due to mechanical instability [3]. Atrophic nonunion occurs when the vascularization at the bone ends has been altered, and typically there is no evidence of callus formation upon X-ray [5]. This presentation typically indicates a problem with the body’s regenerative processes. Pseudarthrosis can be characterized as atrophic or hypertrophic but will typically lead to the formation of a false joint over time [6]. Oligotrophic nonunions exhibit a combination of the characteristics of the three previous nonunion types. Studies have shown that in cases of atrophic nonunion, BMPs are expressed yet the cellular environment is disrupted [7], and fibrogenic collagen III is secreted to provide structure and stability in response to a lack of adequate bone regeneration [8]. The hallmark of atrophic nonunion is the formation of scar tissue [9,10], or fibrosis, which may be referred as scarful healing. The goal of a successful therapeutic strategy to treat atrophic nonunion fractures must be to prevent or divert the scarful healing to a scarless healing.

  1. Andrzejowski, P.; Giannoudis, P.V. The 'diamond concept' for long bone non-union management. J Orthop Traumatol 2019, 20, 21, doi:10.1186/s10195-019-0528-0.
  2. Schlundt, C.; Bucher, C.H.; Tsitsilonis, S.; Schell, H.; Duda, G.N.; Schmidt-Bleek, K. Clinical and Research Approaches to Treat Non-union Fracture. Curr Osteoporos Rep 2018, 16, 155-168, doi:10.1007/s11914-018-0432-1.
  3. Schwabe, P.; Simon, P.; Kronbach, Z.; Schmidmaier, G.; Wildemann, B. A pilot study investigating the histology and growth factor content of human non-union tissue. Int Orthop 2014, 38, 2623-2629, doi:10.1007/s00264-014-2496-6.
  4. Frolke, J.P.; Patka, P. Definition and classification of fracture non-unions. Injury 2007, 38 Suppl 2, S19-22, doi:10.1016/s0020-1383(07)80005-2.
  5. Loi, F.; Cordova, L.A.; Pajarinen, J.; Lin, T.H.; Yao, Z.; Goodman, S.B. Inflammation, fracture and bone repair. Bone 2016, 86, 119-130, doi:10.1016/j.bone.2016.02.020.
  6. Lawton, D.M.; Andrew, J.G.; Marsh, D.R.; Hoyland, J.A.; Freemont, A.J. Mature osteoblasts in human non-union fractures express collagen type III. Mol Pathol 1997, 50, 194-197, doi:10.1136/mp.50.4.194.
  7. Wang, L.; Tower, R.J.; Chandra, A.; Yao, L.; Tong, W.; Xiong, Z.; Tang, K.; Zhang, Y.; Liu, X.S.; Boerckel, J.D.; et al. Periosteal Mesenchymal Progenitor Dysfunction and Extraskeletally-Derived Fibrosis Contribute to Atrophic Fracture Nonunion. J Bone Miner Res 2019, 34, 520-532, doi:10.1002/jbmr.3626.
  8. Rodriguez-Merchan, E.C.; Forriol, F. Nonunion: general principles and experimental data. Clin Orthop Relat Res 2004, 4-12.

  • Materials and Methods: An illustrative image of the randomization of the groups could be made as well as the experimental design.

Response: We appreciate your feedback. We hope that the addition to our Abstract and the editing of both our Materials and Methods and Results sections have helped to clarify our experimental design. However, we do not see the value in illustrating the randomization of the mice because we randomized within the individual mice populations. As such, the groups of mice receiving PBS and PTH have an identical genetic background, age, etc.

  • Materials and Methods: Why was the 5 mm calvaria defect performed and not the 8 mm?

Response: We appreciate you posing this question. An 8 mm calvaria defect is primarily used in the rat model, while the 5 mm defect has been shown to be a nonunion defect in mice. The 5 mm defect is the most commonly used size in the mouse model. Since other readers might have the same question, we have added the following sentence and references to provide support for our decision.

Line 174-175: This defect size was chosen because it is the most commonly used in a mouse model [34-36].

  1. Cooper, G.M.; Mooney, M.P.; Gosain, A.K.; Campbell, P.G.; Losee, J.E.; Huard, J. Testing the critical size in calvarial bone defects: revisiting the concept of a critical-size defect. Plast Reconstr Surg 2010, 125, 1685-1692, doi:10.1097/PRS.0b013e3181cb63a3.
  2. Hermenean, A.; Codreanu, A.; Herman, H.; Balta, C.; Rosu, M.; Mihali, C.V.; Ivan, A.; Dinescu, S.; Ionita, M.; Costache, M. Chitosan-Graphene Oxide 3D scaffolds as Promising Tools for Bone Regeneration in Critical-Size Mouse Calvarial Defects. Sci Rep 2017, 7, 16641, doi:10.1038/s41598-017-16599-5.
  3. Rahman, C.V.; Ben-David, D.; Dhillon, A.; Kuhn, G.; Gould, T.W.; Muller, R.; Rose, F.R.; Shakesheff, K.M.; Livne, E. Controlled release of BMP-2 from a sintered polymer scaffold enhances bone repair in a mouse calvarial defect model. J Tissue Eng Regen Med 2014, 8, 59-66, doi:10.1002/term.1497.

  • Materials and Methods: Does the bone repair process in rats normally occur in seven weeks, why was the four weeks period used in the research? I suggest a justification.

Response: The 4-week timepoint was chosen based on our previous study from 2013. In this study we compared wild-type mice given PTH or PBS and found by longitudinal in vivo µCT that four weeks was the first time point in which significant bone mass was measured in the animals given PTH. In other words, this timepoint is specifically tailored to the PTH regenerative kinetics in the context of a calvarial defect in mice, and thus an appropriate time point for this current study. To provide additional clarity we have added the following statement and re-cited our 2013 paper in the Materials and Methods section.

Line 202-203: a time point chosen based on our previous research in a mouse calvarial defect model [22].

  1. Sheyn, D.; Cohn Yakubovich, D.; Kallai, I.; Su, S.; Da, X.; Pelled, G.; Tawackoli, W.; Cook-Weins, G.; Schwarz, E.M.; Gazit, D.; et al. PTH promotes allograft integration in a calvarial bone defect. Mol Pharm 2013, 10, 4462-4471, doi:10.1021/mp400292p.

  • Results: In figure 1 I suggest making two plates separating the images from the graphics for better visibility and quality. I suggest increasing the size of the graphics of all figures as well as the quality to facilitate and improve the visualization.

Response: We appreciate this comment regarding our figures. We have increased the size of the graphics in hopes that this will increase the clarity.

  • Results: Results can be better described, are confusing and very succinct.

Response: We greatly appreciate this comment and agree that the Results section could have been structured in a more logical way. As such, we have decided to revise the entire Results section for added clarity (Line 231-295).

  • Discussion: The discussion could be written according to and in the same order as the data presented in the results, including the limitations of the study.

Response: The relevant portion of the discussion was reorganized to match the order of data presentation per this suggestion, thank you for bringing this to our attention. We also have added the following sentences in the Discussion section to underscore the limitations of our study.

Line 398-407: A limitation of our experimental design is the fact that there is a difference between the constitutive activation of PTHR signaling and the intermittent administration of PTH used in the clinical setting. Constitutive PTHR activation has been found to have a different effect on osteogenesis [27], when compared to intermittent PTH administration, and at this point the differing effect on angiogenesis has not been fully characterized [26]. In addition, we are unable to directly measure the extent of which PTHR signaling has been ac-tivated, which inhibits our ability to estimate a comparable “dose” of PTH. These limitations present challenges to translating this data to the clinical setting, but we believe that this study provides additional support for the continued research into the use of PTH for allograft-based fracture healing.

  1. Calvi, L.M.; Sims, N.A.; Hunzelman, J.L.; Knight, M.C.; Giovannetti, A.; Saxton, J.M.; Kronenberg, H.M.; Baron, R.; Schipani, E. Activated parathyroid hormone/parathyroid hormone-related protein receptor in osteoblastic cells differentially affects cortical and trabecular bone. J Clin Invest 2001, 107, 277-286, doi:10.1172/JCI11296.
  2. Roche, B.; Vanden-Bossche, A.; Malaval, L.; Normand, M.; Jannot, M.; Chaux, R.; Vico, L.; Lafage-Proust, M.H. Parathyroid hormone 1-84 targets bone vascular structure and perfusion in mice: impacts of its administration regimen and of ovariectomy. J Bone Miner Res 2014, 29, 1608-1618, doi:10.1002/jbmr.2191.

  • Conclusion: Change the conclusion into final considerations, including the purpose of the research, its main results, the conclusions and future clinical perspectives.

Response: The conclusion section was critically revised to include the purpose of the research, main results, conclusions, and future clinical perspective per the reviewer’s suggestion (Line 409-431).

Thank you again to the reviewers for taking the time to review our manuscript.

We look forward to hearing from you in due time regarding our submission and will be happy to respond to any further questions and comments you may have.

Round 2

Reviewer 4 Report

No comments